# Affiliate Stigma in Caregivers of Children with Attention-Deficit/Hyperactivity Disorder: The Roles of Stress-Coping Orientations and Parental Child-Rearing Styles

**DOI:** 10.3390/ijerph18179004

**Published:** 2021-08-26

**Authors:** Chih-Cheng Chang, Yu-Min Chen, Ray C. Hsiao, Wen-Jiun Chou, Cheng-Fang Yen

**Affiliations:** 1Chi Mei Medical Center, Department of Psychiatry, Tainan 70246, Taiwan; rabiata@mail.chimei.org.tw; 2Department of Health Psychology, College of Health Sciences, Chang Jung Christian University, Tainan 71101, Taiwan; 3Department of Psychiatry, Kaohsiung Medical University Hospital, Kaohsiung 80708, Taiwan; bluepooh79@msn.com; 4Department of Psychiatry, School of Medicine, and Graduate Institute of Medicine, Kaohsiung Medical University, Kaohsiung 80708, Taiwan; 5Department of Psychiatry and Behavioral Sciences, University of Washington School of Medicine, Seattle, WA 98295, USA; rhsiao@u.washington.edu; 6Children’s Hospital and Regional Medical Center, Department of Psychiatry, Seattle, WA 98105, USA; 7College of Medicine, Chang Gung University, Taoyuan 33302, Taiwan; 8Kaohsiung Medical Center, Department of Child and Adolescent Psychiatry, Chang Gung Memorial Hospital, Kaohsiung 83301, Taiwan; 9College of Professional Studies, National Pingtung University of Science and Technology, Pingtung 91201, Taiwan

**Keywords:** attention-deficit/hyperactivity disorder, affiliate stigma, coping, child-rearing style

## Abstract

Affiliate stigma may increase the risks of negative parenting and psychological and depressive problems in caregivers of children with attention-deficit/hyperactivity disorder (ADHD). Evaluating affiliate stigma and determining how to reduce it are crucial to promoting mental health in caregivers and their children with ADHD. The aim of this study was to examine the associations of stress-coping orientations and parental child-rearing styles with the risk of high affiliate stigma in caregivers of children with ADHD in Taiwan. Affiliate stigma, stress-coping orientations, and parental child-rearing styles were assessed. The results of univariate logistic regression analysis indicated that caregivers’ gender, depressive symptoms, four orientations of stress coping, and two parenting styles, and children’s high severities of internalizing, externalizing, and ADHD symptoms were significantly associated with high affiliate stigma. The results of multivariate logistic regression analysis indicated that after controlling for caregivers’ gender, depressive symptoms, and children’s severity of internalizing, externalizing, and ADHD symptoms, caregivers with high orientation of seeking social support were less likely to have high affiliate stigma than those with low orientation of seeking social support; the caregivers with high care and affection parenting were less likely to have high affiliate stigma than those with low care and affection parenting, whereas the caregivers with high overprotection parenting were more likely to have high affiliate stigma than those with low overprotection parenting. Intervention programs targeting caregiver affiliate stigma must consider various coping orientations and parental child-rearing styles in their approach.

## 1. Introduction

### 1.1. Affiliate Stigma in Caregivers of Children with Attention-Deficit/Hyperactivity Disorder

Affiliate stigma refers to the self-stigma of caregivers of people with mental illnesses; this stigma entails a caregiver perceiving stigmatizing attitudes being directed at them from others and internalizing these negative attitudes [1]. Affiliate stigma is prevalent among caregivers of children with attention-deficit/hyperactivity disorder (ADHD) [2,3,4]. Research has demonstrated that higher affiliate stigma among caregivers of children with ADHD is associated with more negative parenting [4] and increased distress [3] and depression [5] in caregivers. Higher parental affiliate stigma was also significantly associated with children’s poorer social skills and greater aggression [4]. Caregivers of children with ADHD who have high affiliate stigma are more likely to have unfavorable attitudes toward their children’s ADHD diagnosis, pharmacotherapy, behavioral therapy, as well as etiological explanations of ADHD [2]. Evaluating affiliate stigma and determining how to reduce it are crucial in promoting mental health in caregivers and their children with ADHD.

### 1.2. Coping Orientations and Affiliate Stigma

According to the ecological systems theory [6], caregivers’ affiliate stigma may result from the complex interactions among caregivers, children with ADHD, and their environments. Certain characteristics of caregivers (being a woman and having a high education level) and children with ADHD (being a girl and exhibiting severe inattention symptoms) predict a high degree of affiliate stigma [2]. Other individual and caregiver–child interactions relating to affiliate stigma in the caregivers of children with ADHD warrant further study. Caregivers of children with ADHD may experience great stress as a response to people’s stigmatizing attitudes—or the perception thereof—and consequently develop affiliate stigma; therefore, caregivers’ stress-coping orientations may influence the level of affiliate stigma. Coping is defined as the process of managing specific internal and external sources of psychological stress through cognitive or behavioral efforts [7]. Research has identified multiple categories of coping strategies people use to cope with stress situations [8,9,10,11]. Carver and colleagues developed the Coping Orientation to Problems Experienced (COPE) to assess 13 categories of stress-coping strategies, including active coping, planning, suppression of competing activities, restraint coping, seeking of instrumental social support, seeking of emotional social support, positive reinterpretation, acceptance, turning to religion, focusing on and venting of emotions, denial, behavioral disengagement, and mental disengagement [9]. Research has demonstrated that caregivers of children with an intellectual disability and autism spectrum disorder (ASD) may use problem-solving (e.g., talking to health professionals) [12], turning to religion, self-compassion) [12,13,14], and less effective strategies (e.g., social withdrawal) [12,14,15,16] to cope with the impacts of affiliate stigma. Increased stigma was associated with seeking help from religious centers and traditional healers [12]. To date, no study has examined whether caregivers with various stress-coping orientations experience different levels of ADHD-related affiliate stigma.

### 1.3. Parental Child-Rearing Styles and Affiliate Stigma

Parker developed the Parental Bonding Instrument to measure three factors of parental child-rearing, namely care and affection, overprotection, and authoritarianism [17]. A high score on the care and affection subscale reflects affection and warmth, and a low score indicates rejection or indifference. Overprotection reflects overprotective parenting and denial of their adolescents’ psychological autonomy. Finally, authoritarianism represents the degree of authoritarian-style control that parents exert over their adolescents’ behaviors [17,18]. Research has shown that caregivers’ anticipation of stigma and associated feelings of self-blame may lead to overprotection of the children with intellectual disability from potential harm [14]. Research has indicated that mothers of children with ADHD tend to be less affectionate and more overprotective of their children than mothers of children without ADHD [19,20]. Children’s ADHD symptoms and comorbidity and maternal depression and neuroticism are significantly correlated with various parental child-rearing styles [5,20]. The complicated interactions between caregivers and their children with ADHD may impact the mental health of both parties and people’s stigmatizing attitudes directed toward them; therefore, the role of various parental child-rearing styles in affiliate stigma among caregivers of children with ADHD warrants further study.

### 1.4. Aims of the Present Study

The aim of this study was to examine the associations between stress-coping orientations and parental child-rearing styles with affiliate stigma in caregivers of children with ADHD in Taiwan. We hypothesized that various stress-coping orientations and parental child-rearing styles have various associations with affiliate stigma in caregivers of children with ADHD.

## 2. Methods

### 2.1. Participants and Procedure

The procedure of recruiting participants at baseline in the Study on Affiliate stigma in Caregivers of Children with ADHD in Taiwan has been described elsewhere [5]. In brief, caregivers of children aged ≤18 years diagnosed with ADHD according to the Diagnostic and Statistical Manual of Mental Disorders, Fifth Edition [21] were consecutively recruited for this study between June 2019 and April 2020 from the child and adolescent psychiatric outpatient clinics of two medical centers in Kaohsiung, Taiwan. Two child psychiatrists conducted diagnostic interviews with children and caregivers and established ADHD diagnoses based on DSM-5 criteria. Children who had an intellectual disability or autism spectrum disorder with difficulties in communication were excluded. Caregivers who had an intellectual disability, schizophrenia, bipolar disorder, or any cognitive deficits that resulted in significant communication difficulties were also excluded. A total of 412 caregivers of children who received an ADHD diagnosis were invited to participate in the study. Of these, 12 (2.9%) declined to participate. Thus, 400 (97.1%) caregivers participated in the study and were interviewed by research assistants. The institutional review boards of Kaohsiung Medical University (KMUHIRB-SV(I)-20190130; date of approval: 7 May 2019) and Chang Gung Memorial Hospital, Kaohsiung Medical Center (201900432A3; date of approval: 3 May 2019) approved the study.

### 2.2. Measures

#### 2.2.1. Affiliate Stigma Scale

We used the 22-item self-administrated Affiliate Stigma Scale (ASS) [1] to measure caregivers’ internalization of people’s stigmatizing attitudes toward children’s ADHD. The ASS includes the affective domain (seven items; for example, “I feel inferior because one of my children has ADHD”), cognitive domain (seven items; for example, “My reputation is damaged because I have a child with ADHD at home”), and behavioral domain (eight items; for example, “I dare not tell others that I have a child with ADHD”). Each item was rated on a 4-point Likert scale from 1 (strongly disagree) to 4 (strongly agree). High total scores indicated high levels of self-stigma toward their child’s ADHD. The original version exhibited excellent internal consistency (α = 0.94) and satisfactory predictive validity [1]. The ASS also exhibited robust psychometric properties in a Taiwanese sample [22]. In this study, the Cronbach’s α for the total ASS was 0.95. Because the data were non-normally distributed, we defined the total score above and below the median score of 37 in this study as high and low affiliate stigma, respectively.

#### 2.2.2. Coping Orientation to Problems Experienced (COPE)

We used the 53-item self-administrated COPE to assess caregivers’ stress-coping strategies [9]. Every item is rated on a 4-point Likert scale from 1 (usually do not do this at all) to 4 (usually do this a lot). A high total score on a particular scale indicates that the participant is more likely to cope with stress using those particular strategies. Research has determined that the COPE scale has high reliability and validity [9]. The most widely used categories of coping on the COPE are problem-solving (including active coping, planning, suppression of competing activities, and restraint coping), emotional focused (including seeking of instrumental and emotional social support, positive reinterpretation, acceptance, focusing on and venting of emotions, and turning to religion), and less effective strategies (including denial and behavioral and mental disengagement) [9]. The Cronbach’s α of problem-solving (median = 3.125) and less effective coping subscales (median = 1.583) were 0.936 and 0.849, respectively, whereas the Cronbach’s α of the emotion-focused coping subscale was 0.542. Therefore, we further divided the emotion-focused coping subscale into three subscales, including seeking social support (including seeking of instrumental and emotional social support; Cronbach’s α = 0.924; median = 3), changing cognition (including positive reinterpretation and acceptance; Cronbach’s α = 0.884; median = 3.25), and passive emotional coping (including focusing on and venting of emotions and turning to religion; Cronbach’s α = 0.865; median = 2.25). Because the data were non-normally distributed, we defined the total score of each subscale above and below the median score in this study as high and low coping orientation, respectively.

#### 2.2.3. Chinese Version of the Caregiver-Reported Parental Bonding Instrument (PBI)

The original PBI measures how each adult remembers and represents their own parents’ attitudes and behaviors towards them during their infancy [17]. Gau and colleagues transformed the original PBI into the 25-item caregiver-reported version of the PBI to measure caregivers’ attitudes and behaviors toward their young children or children with a developmental disability in the few years preceding the assessment [23]. The Chinese version of the caregiver-reported PBI has been used to assess caregiver-reported bonding with their children with ADHD [19,20,24], children with Down syndrome [25], and children in community [26]. Each item is rated on a 4-point Likert scale from 1 (very likely) to 4 (very unlikely). The PBI contains the following three principal dimensions: care/affection (12 items), overprotection (seven items), and authoritarianism (six items). The reliability and validity of the Chinese PBI have been described elsewhere [23]. The Cronbach’s α for the subscales of parent-reported care and affection, overprotection, and authoritarianism were 0.78, 0.74, and 0.72, respectively. Because the data were non-normally distributed, we defined the total score of each dimension above and below the median score in this study as high and low child-rearing orientation, respectively. The median scores of care/affection, overprotection, and authoritarianism were 39, 14, and 12, respectively.

#### 2.2.4. Child Behavior Checklist for Ages 6–18

The Chinese Version of the Child Behavior Checklist for Ages 6–18 (CBCL/6-18) is a 112-item standardized caregiver-reported measure of behavioral problems in children [27,28,29]. We used the recommended T-score transformations of raw behavior scores, adjusted for age and sex differences, for behavior exhibited in normative samples. Internalizing problems (including anxious and depressed, withdrawn and depressed, and somatic complaint scales), externalizing problems (including rule-breaking and aggressive behavior scales), and ADHD symptoms of the CBCL/6-18 were used for analysis. We defined the T score of each dimension above and below the median T score in this study as high and low symptoms, respectively. The median scores of internalizing problems, externalizing, and ADHD symptoms were 61, 60, and 62, respectively.

#### 2.2.5. Center for Epidemiological Study-Depression Scale

The 20-item Mandarin Chinese version of the Center for Epidemiological Study-Depression Scale (CES-D) was used to assess the severity of depressive symptoms in caregivers [30,31]. Caregivers were asked how often they experienced each depressive symptom in the preceding month. Response categories are (0) rarely or none of the time (less than 1 day), (1) some or a little of the time (l–2 days), (2) occasionally or a moderate amount of the time (3–4 days), or (3) most or all of the time (5–7 days). A higher total score indicates more severe depressive symptoms. Cronbach’s α for the scale in the present study was 0.88. We defined the total score above and below the median score of 12 in this study as high and low depressive symptoms, respectively.

#### 2.2.6. Demographic Characteristics

Caregivers’ gender, age, length of education, and marital status, and children’s gender and age were collected at baseline.

### 2.3. Statistical Analysis

Descriptive results are presented as percentages for categorical variables and as mean and standard deviation (SD) for continuous variables. The associations of caregivers’ demographic characteristics, depressive symptom, high orientation of stress-coping strategies, and child-rearing styles, and children’s demographic characteristics, and severe behavioral and ADHD symptoms (independent variables) with high affiliate stigma (dependent variable) were firstly examined by using univariate logistic regression analysis. Variables that were significantly associated with high affiliate stigma in univariate logistic regression analysis were further selected for multivariate logistic regression analysis, serving as the independent variables. The associations of various coping orientations and parental bonding with high affiliate stigma were examined separately in two logistic regression models, controlling for demographic characteristics, caregivers’ depressive symptoms, and the children’s behavioral and ADHD symptoms. Odds ratio (OR), its 95% confidence interval (CI), and a two-tailed *p* value of <0.05 were used to present statistical significance. We defined the OR that was higher than 1.5, 2.5, and 4 or lower than 0.667, 0.4, and 0.25 as having a small, medium, and large effect, respectively [32,33].

## 3. Results

Table 1 presents the mean, SD, and percentage of caregivers’ affiliate stigma, demographics, depressive symptoms, stress-coping orientation, and child-rearing styles, and children’s demographics, behavioral, and ADHD symptoms. In total, 321 female and 79 male caregivers of children with ADHD participated in the study; most of the children were boys (*n* = 322, 80.5%).

Table 2 presents the results of univariate logistic regression examining the associations of caregivers’ and children’s factors with caregivers’ high affiliate stigma. Regarding caregivers’ factors, the results indicated that male caregivers were less likely to have high affiliate stigma than females. The caregivers with high depressive symptoms and high orientation of less effective coping were more likely to have high affiliate stigma than those with low depressive symptoms and low orientation of less effective coping, respectively. The caregivers with high orientation of problem-focused coping, seeking social support, and changing cognition were less likely to have high affiliate stigma than those with low orientation of these coping strategies, whereas the caregivers with high orientation of less effective coping were more likely to have high affiliate stigma than those with low orientation of less effective coping. The caregivers with high care and affection parenting were less likely to have high affiliate stigma than those with low care and affection parenting. The caregivers with high overprotection and authoritarianism parenting were more likely to have high affiliate stigma than those with low overprotection and authoritarianism parenting, respectively. Regarding children’s factors, caregivers with high internalizing and externalizing problems and ADHD symptoms of children were more likely to have high affiliate stigma than those with low symptoms of children.

Variables that were significantly associated with high affiliate stigma in univariate logistic regression analysis were further selected for multivariate logistic regression analysis. Caregivers’ gender and depressive symptoms and children’s internalizing, externalizing, and ADHD symptoms were used as the covariates. Table 3 details the results of the multivariate logistic regression analysis. The results of Model I examining the role of stress-coping orientations indicated that after controlling for the covariates, caregivers with high orientation of seeking social support were less likely to have high affiliate stigma than those with low orientation of seeking social support. The results of Model II examining the role of parenting styles indicated that caregivers with high care and affection parenting were less likely to have high affiliate stigma than those with low care and affection parenting. The caregivers with high overprotection parenting were more likely to have high affiliate stigma than those with low overprotection parenting. All OR values indicated a small effect.

## 4. Discussion

The study revealed that various types of stress-coping orientations and parental childrearing styles had varying relationships with high affiliate stigma among caregivers of children with ADHD.

### 4.1. Coping Orientations and Affiliate Stigma

Parents may experience multiple sources of stress when taking care of children with ADHD such as managing their children’s behaviors, supervising the completion of their children’s daily routines, maintaining relationships among family members, and craving social support [34]. The types of stress-coping strategies the caregivers adopt may determine the effectiveness of this management, which may further influence caregivers’ self-efficacy in managing their children’s problematic behaviors, children’s coordination with caregivers, and comments from people other than caregivers and children. This study demonstrated that caregivers with high orientation of seeking social support were less likely to have high affiliate stigma than those with low orientation of seeking social support, indicating that seeking social support is an adaptive stress-coping strategy. The present study examined two dimensions of seeking social support, including instrumental and emotional social support. Seeking of instrumental social support such as learning from the experience of others may help the caregivers increase the opportunity to successfully manage stressors that they encounter. Seeking emotional social support such as getting sympathy and understanding from others may diminish the negative emotional consequences of stressful events [8,9]. Social support may contribute to caregivers’ success in managing children’s ADHD-related problems and enhance caregivers’ emotional health. Social support may also reduce the risk of social withdrawal for caregivers of children with ADHD and increase the opportunity for people to understand caregivers’ care burden. Therefore, social support may reduce people’s ADHD-related stigmatizing attitudes toward the caregivers.

In addition to seeking socioemotional support, the coping strategy of changing cognition such as positive reinterpretation, acceptance of stressors, and perceived growth in the process of managing stress may diminish the negative emotional consequences of stressful events [8,9]. Because negative emotions may contribute to people’s misconceptions of social clues in social interactions [35], it is reasonable to hypothesize that caregivers adopting the coping strategy of changing cognition may experience less internalization of ADHD-related stigmatizing attitudes. However, the prediction of changing cognition for low affiliate stigma was found in the univariate but not in the multivariate logistic regression analysis. Adoption of less effective coping strategies such as denial and behavioral and mental disengagement may delay and compromise the effects of managing ADHD-related problems for caregivers. Contrarily, adoption of problem-solving coping strategies may increase the efficiency of managing ADHD-related problems. However, the predictions of less effective and problem-solving coping strategies for affiliate stigma were found in univariate but not in multivariate logistic regression analysis.

### 4.2. Parental Child-Rearing Styles and Affiliate Stigma

The results of this study demonstrated that caregivers with high care and affection parenting were less likely to have high affiliate stigma than those with low care and affection parenting, whereas the caregivers with high overprotection parenting were more likely to have high affiliate stigma than those with low overprotection parenting. Research has demonstrated that high levels of caregiver care and affection significantly correlate with reduced inattention and psychiatric comorbidities in their children [20]. Charbonnier et al. demonstrated that children’s ADHD symptoms were positively associated with affiliate stigma [3]. Therefore, the parental style of care and affection may decrease the risk of high affiliate stigma. However, the present study found that the parental style of care and affection decreased the risk of high affiliate stigma after controlling for the effects of children’s ADHD symptoms. The result indicated that the association between the parental style of care and affection and low affiliate stigma may not be fully accounted for by the reduced ADHD symptoms. It is possible that both the parental style of care and affection and low affiliate stigma resulted from caregivers’ healthy psychological functioning. A happier person may score higher on various kinds of measures reflecting psychological well-being.

Furthermore, overprotective parenting reflects denial of their children’s psychological autonomy [18]. Research has found that ADHD symptoms were significantly correlated with both caregivers’ overprotection [20] and affiliate stigma [2]. It is also possible that both parental child-rearing styles and affiliate stigma are the results of caregivers’ psychological and cognitive mechanisms. The results of this study indicated that intervention programs targeting caregiver affiliate stigma must consider the parental styles in their approach.

### 4.3. Implication

Although the cross-sectional study design limited the possibility to determine the causal relationship between affiliate stigma, stress-coping orientations, and parental child-rearing styles, mental health professionals must assist the caregivers of children with ADHD in identifying the most suitable stress-coping strategies, especially seeking of social support, and advising against less effective coping strategies. Moreover, intervention programs targeting caregiver affiliate stigma must consider parental styles in their approach. Affiliate stigma, stress-coping strategies, and parental child-rearing styles may be the results of interactions between caregivers and environments. Not only caregivers’ psychological and cognitive mechanisms but also the interactions with people in caregivers’ microsystem and mesosystem should be assessed.

### 4.4. Limitations

This study surveyed the association between affiliate stigma and stress-coping orientations and parental child-rearing styles by a cross-sectional design, which limited the possibility of determining the causal relationships among the variables. Moreover, this study collected the data from caregivers but no other information source. The single data source may have resulted in common-method variance. This study did not collect data on children’s treatment for ADHD and thus could not determine the possible influence of ADHD treatment. Further study is needed to support the categorization of the COPE used in this study.

## 5. Conclusions

This study demonstrated that seeking social support may be an adaptive stress-coping strategy to protect caregivers of children with ADHD from aggravated affiliate stigma. Moreover, adopting the parental style of care and affection may contribute to low affiliate stigma, whereas adopting the parental style of overprotection may increase the risk for caregivers perceiving public stigma and internalizing the negative attitudes. The results of this study indicate that caregivers of children with ADHD need help to develop the coping strategy of seeking social support. Intervention programs targeting caregiver affiliate stigma must consider parental styles in their approach. Further study is needed to examine the mechanism accounting for the associations of seeking social support and the parental style of care and affection and overprotection with affiliate stigma in caregivers of children with ADHD. Mental health professionals should develop the intervention programs to enhance caregivers’ adaptive stress-coping strategies and parental child-rearing styles; the effects of intervention programs on reducing affiliate stigma also warrant study.

## Figures and Tables

**Table 1 ijerph-18-09004-t001:** Caregivers’ affiliate stigma, demographics, depressive symptoms, stress-coping orientation, and child-rearing styles, and children’s demographics, behavioral, and ADHD symptoms (*n* = 400).

Variable	*n* (%)	Mean (SD)	Range
*Caregivers’ characteristics*			
Affiliate stigma			
Low	212 (53.0)		
High	188 (47.0)		
Gender			
Female	321 (80.3)		
Male	79 (19.8)		
Age (years)		43.2 (7.0)	23–69
Length of education (years)		14.2 (3.2)	0–28
Marriage status			
Married or cohabited	319 (79.8)		
Separated or divorced	81 (20.3)		
Depressive symptoms			
Low	208 (52.0)		
High	192 (48.0)		
Problem-focused coping			
Low	201 (50.3)		
High	199 (49.8)		
Seeking social support			
Low	214 (53.5)		
High	186 (46.5)		
Less effective coping			
Low	227 (56.7)		
High	173 (43.3)		
Changing cognition			
Low	224 (56.0)		
High	176 (44.0)		
Passive emotional coping			
Low	222 (55.5)		
High	178 (44.5)		
Care/affection			
Low	218 (54.5)		
High	182 (45.5)		
Overprotection			
Low	242 (60.5)		
High	158 (39.5)		
Authoritarianism			
Low	231 (57.8)		
High	169 (42.3)		
*Children’s characteristics*			
Gender			
Girl	78 (19.5)		
Boy	322 (80.5)		
Age (years)		10.3 (3.2)	6–18
Internalizing problems			
Low	210 (52.5)		
High	190 (47.5)		
Externalizing problems			
Low	211 (52.8)		
High	189 (47.3)		
ADHD symptoms			
Low	206 (51.5)		
High	194 (48.5)		

**Table 2 ijerph-18-09004-t002:** Factors related to high affiliate stigma in caregivers of children with ADHD: Univariate logistic regression.

Variable	High Affiliate Stigma
	OR (95% CI)
*Caregiver*	
Age	1.015 (0.987–1.044)
Male ^a^	0.591 (0.356–0.981) *
Marriage status of separation or divorce ^b^	1.445 (0.886–2.358)
Length of education	0.981 (0.922–1.045)
High depressive symptoms ^c^	3.431 (2.274–5.178) ***
High problem-focused coping ^d^	0.471 (0.316–0.703) ***
High seeking social support ^d^	0.365 (0.243–0.548) ***
High changing cognition ^d^	0.388 (0.258–0.584) ***
High passive emotional coping ^d^	0.974 (0.656–1.445)
High less effective coping ^d^	1.989 (1.331–2.972) **
High care/affection ^e^	0.346 (0.229–0.521) ***
High overprotection ^e^	2.525 (1.673–3.810) ***
High authoritarianism ^e^	1.546 (1.037–2.305) *
*Children’s characteristics*	
Boy ^f^	0.861 (0.525–1.413)
Age	1.018 (0.957–1.083)
High internalizing problems ^g^	2.532 (1.691–3.790) ***
High externalizing problems ^g^	2.584 (1.725–3.870) ***
High ADHD symptoms ^g^	1.822 (1.224–2.710) **

ADHD: attention-deficit/hyperactivity. ^a^ Female as the reference; ^b^ marriage status of being married or cohabited as the reference; ^c^ low depressive symptoms as the reference; ^d^ low orientation of coping as the reference; ^e^ low adoption of child-rearing style as the reference; ^f^ girl child as the reference; ^g^ low symptoms as the reference. ***: *p* < 0.05; ****: *p* < 0.01; *****: *p* < 0.001.

**Table 3 ijerph-18-09004-t003:** Factors related to high affiliate stigma in caregivers of children with ADHD: Multivariate logistic regression.

Variable	High Affiliate Stigma	
	Model I OR (95% CI)	Model II OR (95% CI)
Male caregivers ^a^	0.577 (0.325–1.023)	0.665 (0.376–1.178)
Caregivers’ high depressive symptoms ^b^	2.172 (1.367–3.451) ***	2.093 (1.328–3.301) **
Children’s high internalizing problems ^c^	0.512 (0.321–0.818) **	1.498 (0.929–2.418)
Children’s high externalizing problems ^c^	1.585 (0.994–2.527)	1.308 (0.775–2.210)
Children’s high ADHD symptoms ^c^	1.060 (0.670–1.676)	1.178 (0.716–1.938)
High problem-focused coping ^d^	0.866 (0.530–1.413)	
High seeking social support ^d^	0.505 (0.309–0.825) **	
High changing cognition ^d^	0.659 (0.404–1.074)	
High less effective coping ^d^	1.369 (0.866–2.165)	
High care/affection ^e^		0.471 (0.297–0.748) **
High overprotection ^e^		1.678 (1.062–2.651) *
High authoritarianism ^e^		1.004 (0.639–1.578)

ADHD: attention-deficit/hyperactivity. ^a^ Female as the reference; ^b^ low depressive symptoms as the reference; ^c^ low symptoms as the reference; ^d^ low orientation of coping as the reference; ^e^ low adoption of child-rearing style as the reference. ***: *p* < 0.05; ****: *p* < 0.01; *****: *p* < 0.001.

## Data Availability

The data will be available upon reasonable request to the corresponding authors.

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
