# Peer review of "Affiliate Stigma in Caregivers of Children with Attention-Deficit/Hyperactivity Disorder: The Roles of Stress-Coping Orientations and Parental Child-Rearing Styles"

_ijerph, 2021, doi:10.3390/ijerph18179004_

Round 1

Reviewer 1 Report

The  manuscript addressed the quite interesting issues regarding affiliate stigma of parents of ADHD. The authors found that stress-coping styles and rearing styles are associated with stigma.    Specifically, authors showed emotion-focused coping was negatively associated with stigma, the parental style of care and affection was negatively associated with it,  and the parental style of overprotection was positively associated with it. The explanation on association results of each coping style or rearing style and stigma is described in the discussion section but seems somewhat lacking in reasonable explanation.  The lack of controlling or consideration on various other factors  (socioeconomic status, comorbidities (both parent and child), taking medication or not, and so on ) is also thought to be a major limitation of this manuscript. 

minor

  1. Abstract; The background of the study need to be addressed before the aim of the study. The measures that were used in the study should be addressed more concisely, and the method of statistical analysis should be more defined. 

2. Please present the characteristics (clinical or demographic) of children. Because the severity of the ADHD symptoms or whether they are taking medications may have an impact on the coping or rearing style. If it's not possible to present, it should be mentioned as a limitation.

Author Response

We appreciated your valuable comments. As discussed below, we have revised our manuscript based on your comments. Please let us know if we need to provide anything else regarding this revision.

Comment 1

The explanation on association results of each coping style or rearing style and stigma is described in the discussion section but seems somewhat lacking in reasonable explanation.

Response

Thank you for your comment. We revised the contents of Discussion section regarding the association between affiliate stigma, stress-coping strategies, and parental child-rearing styles as below.

This study demonstrated that high orientation of seeking social support significantly decreased the risk of high ADHD-related affiliate stigma. The present study examined two dimensions of seeking social support, including instrumental and emotional social support. Seeking of instrumental social support such as learning from the experience of others may help the caregivers increases the opportunity to successfully manage stressors that they encounter. Seeking of emotional social support such as getting sympathy and understanding from others may diminish the negative emotional consequences of stressful events [8,9]. Social support may contribute to caregivers’ success in managing children’s ADHD-related problems and enhance caregivers’ emotional health. Social support may also reduce the risk of social withdrawal for caregivers of children with ADHD and increase the opportunity for people to understand caregivers’ care burden. Therefore, social support may reduce people’s ADHD-related stigmatizing attitudes toward the caregivers.

In addition to seeking socioemotional support, the coping strategy of changing cognition such as positive reinterpretation, acceptance of stressors, and perceived growth in the process of managing stress may diminish the negative emotional consequences of stressful events [8,9]. Because negative emotions may contribute to people’s misconceptions of social clues in social interactions [28], it is reasonable to hypothesize that caregivers adopting the coping strategy of changing cognition may experience less internalization of ADHD-related stigmatizing attitudes.表單的底部 However, the prediction of changing cognition for low affiliate stigma was found in univariate but not in multivariate logistic regression analysis. Adoption of less effective coping strategies such as denial and behavioral and mental disengagement may delay and compromise the effects of managing ADHD-related problems for caregivers. Contrarily, adoption of problem-solving coping strategies may increase the efficiency of managing ADHD-related problems. However, the predictions of less effective and problem-solving coping strategies for affiliate stigma were found in univariate but not in multivariate logistic regression analysis. Please refer to line 276-305.

The results of this study demonstrated that the parental style of care and affection decreased the risk of high affiliate stigma, whereas the overprotective style increased the risk of high affiliate stigma. Research has demonstrated that high levels of caregiver care and affection significantly correlate with reduced inattention and psychiatric comorbidities in their children [20]. Charbonnier et al. demonstrated that children’s ADHD symptoms were positively associated with affiliate stigma [3]. Therefore, the parental style of care and affection may decrease the risk of high affiliate stigma. However, the present study found that the parental style of care and affection decreased the risk of high affiliate stigma after controlling for the effects of children’s ADHD symptoms. The result indicated that the association between the parental style of care and affection and low affiliate stigma may not be fully accounted for the reduced ADHD symptoms. It is possible that both the parental style of care and affection and low affiliate stigma resulted from caregivers’ healthy psychological functioning. A happier person may score higher on various kinds of measures reflecting psychological well-being.

Furthermore, overprotective parenting reflects denial of their children’s psychological autonomy [18]. Research has found that ADHD symptoms were significantly correlated with both caregivers’ overprotection [20] and affiliate stigma [2]. It is also possible that both parental child-rearing styles and affiliate stigma are the results of caregivers’ psychological and cognitive mechanisms. The results of this study indicated that intervention programs targeting caregiver affiliate stigma must consider the parental styles in their approach.Please refer to line 307-328.

Comment 2

The lack of controlling or consideration on various other factors (socioeconomic status, comorbidities (both parent and child), taking medication or not, and so on) is also thought to be a major limitation of this manuscript. 

Response

Thank you for your reminding.

  1. We added caregivers’ length of education and marriage status as below into the revised manuscript to indicate caregivers’ socioeconomic status. Please refer to line 201-203.

2.2.6. Demographic Characteristics

Caregivers’ gender, age, length of education and marital status, and children’s gender and age were collected at baseline.”

  1. We also added caregivers’ depressive symptoms into the revised manuscript. Please refer to line 190-200.

2.2.5. Center for Epidemiological Study-Depression Scale

The 20-item Mandarin Chinese version of the Center for Epidemiological Study-Depression Scale (CES-D) was used to assess the severity of depressive symptoms in caregivers [27,28]. Caregivers were asked how often they experienced each depressive symptom in the preceding month. Response categories are (0) rarely or none of the time (less than 1 day), (1) some or a little of the time (l–2 days), (2) occasionally or a moderate amount of the time (3–4 days), or (3) most or all of the time (5–7 days). A higher total score indicates more severe depressive symptoms. Cronbach’s α for the scale in the present study was 0.88. We defined the total score above and below the median score of 12 in this study as high and low depressive symptoms, respectively.

  1. Although we did not assess children’s comorbidities, we used children’s internalizing and externalizing symptoms on the CBCL/6-18 to indicate their psychopathology in addition to ADHD symptoms.
  2. We did not collect the data of children’s treatment for ADHD. We added it as one of limitations. Please refer to line 346-348.

This study did not collect the data of children’s treatment for ADHD and thus could not determine the possible influence of ADHD treatment.

Comment 3

Abstract; The background of the study need to be addressed before the aim of the study. The measures that were used in the study should be addressed more concisely. The method of statistical analysis should be more defined.

Response

Thank you for your comment. We added the background into Abstract. We also revised the methods in Abstract. Please refer to line 23-26 and 28-31.

Affiliate stigma may increase the risks of negative parenting, psychological and depressive problems in in caregivers of children with attention-deficit/hyperactivity disorder (ADHD). Evaluating affiliate stigma and determining how to reduce it are crucial to promoting mental health in caregivers and their children with ADHD. …Affiliate stigma, stress-coping orientations, and parental child-rearing styles were assessed. The predictions of various coping orientations and parental child-rearing styles for affiliate stigma were examined by univariate and multivariate logistic regression analysis.

Comment 4

Please present the characteristics (clinical or demographic) of children. Because the severity of the ADHD symptoms or whether they are taking medications may have an impact on the coping or rearing style. If it's not possible to present, it should be mentioned as a limitation.

Response

  1. We presented children’s gender, age, severities of internalizing, externalizing, and ADHD symptoms in the manuscript. Please refer to Tables 1 and 2.
  2. As described in the response to Comment 2, we did not collect the data of children’s treatment for ADHD. We added it as one of limitations. Please refer to line 346-348.

This study did not collect the data of children’s treatment for ADHD and thus could not determine the possible influence of ADHD treatment.

Reviewer 2 Report

This is a very well written study of the association of parental child-rearing styles and coping strategies with affiliate stigma in a large sample of parents of youth with ADHD. It addresses an important gap in the literature and provides new insights into parent-child interactions in the field of stigma research being relevant for care providers. I only have very minor comments authors may want to implement, as shown below.

  • Introduction: It does not seem the authors use variables of parental well-being, such as maternal depression etcetera. Authors may remove those remarks from the introduction since this is not being investigated in the present study and may guide the reader in another direction. (“Children’s ADHD symptoms and comorbidity and maternal depression and neuroticism are significantly correlated with various parental child-rearing styles [5,20].”)

  • Section 1.1. “It is also associated with increased behavioral problems in children, which manifest as a lack of social skills and aggression [4].” Do refer authors to ADHD or to affiliate stigma by “it”? Also, referring to social skills is a bit confusing, behavioral problems are generally defined as problems in the externalizing domain, while social skills are more autistic-type of problems. Please rephrase.

  • Section 1.1. The sentence of “The most widely used categories of coping are problem-solving, emotional focused, and less effective strategies [8-11].” could be better phrased. Better refer to “emotional focused strategies” for example. Also, I assume emotional strategies might as well be less effective or maladaptive (e.g., emotional responding). Can you clarify?

  • Section 1.2. Small typo, should be “Research has”.

  • Section 2.1. Although the sample is described elsewhere, it would be helpful to know about more about this sample or study. For example, when and how did data collection take place?

  • Section 2.2.4 I am unclear about what authors mean by the section “We determined” until the end of the paragraph. Were authors looking for an internalizing symptom scale? Please clarify in the paper what authors mean by “behavioral problems”.

  • Section 2.2. How exactly was the severity of ADHD symptoms assessed? How caregivers’ education?

  • Section 2.3. Can authors provide more details about the distribution of data, outliers, missing values, possible data transformation, etcetera? I am not sure what authors refer to as ‘behavioral data’ (internalizing problems)? Can authors please clarify this?

  • Results section: Can authors comment about effect sizes and indicate those in the statistical analysis section 2.3? Can authors also please add the effect sizes belonging to the regression analyses in Table 2? Moreover, the effect of emotion-focused coping would not significant anymore after Bonferroni correction for multiple testing, authors could note this, since the sample size was not that small leaving enough power. Clearly, the effect for ‘less effective coping’ was more convincing. Moreover, it is worthwhile to mention that effects remained significant independent from internalizing and ADHD problem severity while adjusting for it in the multivariate models.

  • Discussion, 4. First sentence, please refer to ‘parenting styles’. First sentence under 4.1, the phrasing could be easier, e.g. relate it to lower stigma and higher stigma (rather than saying negatively and positively related); perhaps address which is adaptive and which non-adaptive coping. I see that examples are following. However, emotion focused coping can be positive or negative. As such, I do not understand authors’ reasonings why caregivers may experience fewer negative emotions, can authors please address this (“Because negative emotions may contribute to people’s misconceptions of social clues in social interactions [28], caregivers adopting emotion-focused coping strategies may experience fewer negative emotions, perceive less ADHD-related stigmatizing attitudes being directed toward them, and experience less internalization of these negative attitudes.”).

  • Section 4.1. latter part, would it not be advisable to summarize a separate section addressing possible clinical implications at the end of the discussion.

  • Section 4.2. Here, in the first sentence, authors repeat themselves again, again it could be phrased more attractively and less technical as in a results section. For example, “As may be expected, a positive parental style of care and affection was associated with experiencing less affiliate stigma….” Etcetera. In contrast, overprotection usually refers to a maladaptive or negative parenting style.

  • Regarding “Research has demonstrated that high levels of caregiver care and affection significantly correlate with reduced inattention and psychiatric comorbidities in their children [20], which may in turn reduce other people’s negative attitudes toward caregivers.”, on page 8, it may be worthwhile to mention that the reported association was independent from ADHD symptom severity. All in all, authors may also mention that a generally more healthy psychological functioning may be expressed in all kinds of better psychological measures to which affiliate stigma may also belong, in other words, a happier person will score better on different kinds of measures reflecting psychological well-being, and vice versa, a person with more maladaptive coping and parenting strategies will also be more susceptible to experiencing affiliate stigma.

  • Section 4.2, I assume the authors refer to their own study? Then better say: “the results of this study indicate”

  • Conclusions: Again here, better avoid too technical language. It reads much like a reiteration of results. Wording is quite repetitive. Further, what suggestions for further research can authors give us?

Author Response

Comment 1

Introduction: It does not seem the authors use variables of parental well-being, such as maternal depression etcetera. Authors may remove those remarks from the introduction since this is not being investigated in the present study and may guide the reader in another direction. (“Children’s ADHD symptoms and comorbidity and maternal depression and neuroticism are significantly correlated with various parental child-rearing styles [5,20].”)

Response

Thank you for your reminding. We added caregivers’ depressive symptoms into analysis as a covariate in the revised manuscript. We added the method of survey caregivers’ depressive symptoms as below into Methods section. Please refer to line 190-200. We also added the variable of depressive symptoms into tables.

2.2.5. Center for Epidemiological Study-Depression Scale

The 20-item Mandarin Chinese version of the Center for Epidemiological Study-Depression Scale (CES-D) was used to assess the severity of depressive symptoms in caregivers [27,28]. Caregivers were asked how often they experienced each depressive symptom in the preceding month. Response categories are (0) rarely or none of the time (less than 1 day), (1) some or a little of the time (l–2 days), (2) occasionally or a moderate amount of the time (3–4 days), or (3) most or all of the time (5–7 days). A higher total score indicates more severe depressive symptoms. Cronbach’s α for the scale in the present study was 0.88. We defined the total score above and below the median score of 12 in this study as high and low depressive symptoms, respectively.

Comment 2

Section 1.1. “It is also associated with increased behavioral problems in children, which manifest as a lack of social skills and aggression [4].” Do refer authors to ADHD or to affiliate stigma by “it”? Also, referring to social skills is a bit confusing, behavioral problems are generally defined as problems in the externalizing domain, while social skills are more autistic-type of problems. Please rephrase.

Response

Thank you for your suggestion. We rephrased this sentence as below. Please refer to line 50-51.

Higher parental affiliate stigma was also significantly associated with children's poorer social skills and greater aggression [4].

Comment 3

Section 1.1. The sentence of “The most widely used categories of coping are problem-solving, emotional focused, and less effective strategies [8-11].” could be better phrased. Better refer to “emotional focused strategies” for example. Also, I assume emotional strategies might as well be less effective or maladaptive (e.g., emotional responding). Can you clarify?

Response

Thank you for comment. We reviewed the literatures and found that the results of previous on categories of stress-coping strategies varied. Especially, a portion of “emotional focused strategies” are positive and help solve problems, whereas a portion of “emotional focused strategies” are less effective or maladaptive. Therefore, we replaced the categories of problem-solving, emotional focus, and less effective strategies by the categories developed by Carver, Scheier and Weintraub (1989) as below in Introduction section of the revised manuscript.

“Carver and colleagues developed the Coping Orientation to Problems Experienced (COPE) to assess 13 categories of stress-coping strategies, including active coping, planning, suppression of competing activities, restraint coping, seeking of instrumental social support, seeking of emotional social support, positive reinterpretation, acceptance, turning to religion, focusing on and venting of emotions, denial, behavioral disengagement, and mental disengagement [9].” Please refer to line 71-76.

Comment 4

Section 1.2. Small typo, should be “Research has”.

Response

Thank you for your reminding. We revised it into “Research has”. Please refer to line XXX.

Comment 5

Section 2.1. Although the sample is described elsewhere, it would be helpful to know about more about this sample or study. For example, when and how did data collection take place?

Response

We added the introduction for the method of recruiting participants at baseline as below. Please refer to line 114-126.

In brief, caregivers of children aged ≤18 years diagnosed with ADHD according to the Diagnostic and Statistical Manual of Mental Disorders, Fifth Edition [21] were consecutively recruited for this study between June 2019 and April 2020 from the child and adolescent psychiatric outpatient clinics of two medical centers in Kaohsiung, Taiwan. Two child psychiatrists conducted diagnostic interviews with children and caregivers and established ADHD diagnoses based on DSM-5 criteria. Children who had an intellectual disability or autism spectrum disorder with difficulties in communication were excluded. Caregivers who had an intellectual disability, schizophrenia, bipolar disorder, or any cognitive deficits that resulted in significant communication difficulties were also excluded. A total of 412 caregivers of children who received an ADHD diagnosis were invited to participate in the study. Of these, 12 (2.9%) declined to participate. Thus, 400 (97.1%) caregivers participated in the study and were interviewed by research assistants.

Comment 6

Section 2.2.4 I am unclear about what authors mean by the section “We determined” until the end of the paragraph. Were authors looking for an internalizing symptom scale? Please clarify in the paper what authors mean by “behavioral problems”.

Response

Thank you for your comment. We revised Section 2.2.4. Child Behavior Checklist For Ages 6–18 to define behavioral problems as below. Please refer to line 181-186.

The Chinese Version of the Child Behavior Checklist for Ages 6–18 (CBCL/6-18) is a 112-item standardized caregiver-reported measure of behavioral problems in children [24-26]. We used the recommended T-score transformations of raw behavior scores, adjusted for age and sex differences, for behavior exhibited in normative samples. Internalizing problems (including anxious and depressed, withdrawn and depressed, and somatic complaint scales), externalizing problems (including rule-breaking and aggressive behavior scales), and ADHD symptoms of the CBCL/6-18 were used for analysis.

Comment 7

Section 2.2. How exactly was the severity of ADHD symptoms assessed? How caregivers’ education?

Response

We assessed caregiver-reported ADHD using the CBCL/6-18. Please refer to line 185-186. We added a new paragraph to introduce caregivers’ and children’s demographic characteristics including caregivers’ length of education as the covariates of logistic regression analysis as below. Please refer to line 201-203.

2.2.6. Demographic Characteristics

Caregivers’ gender, age, length of education and marital status, and children’s gender and age were collected at baseline.

Comment 8

Section 2.3. Can authors provide more details about the distribution of data, outliers, missing values, possible data transformation, etcetera? I am not sure what authors refer to as ‘behavioral data’ (internalizing problems)? Can authors please clarify this?

Response

Thank you for your suggestion. We used Shapiro-Wilk test to examine the normality of continuous variables, including affiliate stigma, stress-coping orientation, parental child-rearing styles, and behavioral and ADHD symptoms and found that they were not normally distributed. Therefore, we transformed these continuous variables into dichotomous variables by using the correspondent cutoffs for each instrument. We revised the contents of Methods section regarding the introduction of the measures as below. Because the dependent variable (affiliate stigma) was transformed into the dichotomous variable, the method of statistical analysis was changed to logistic regression analysis. We also added the proportions of participants with various affiliate stigma, stress-coping orientation, child-rearing styles, and symptoms in Table 1. The term “behavioral data” was removed from Table 1.

2.2.1. Affiliate Stigma Scale “...Because that the data was non-normally distributed, we defined the total score above and below the median score of 37 in this study as high and low affiliate stigma, respectively. Please refer to line 144-146.

2.2.2. Coping Orientation to Problems Experienced (COPE) “…Because that the data was non-normally distributed, we defined the total score of each subscale above and below the median score in this study as high and low coping orientation, respectively.” Please refer to line 161-164.

2.2.3. Chinese Version of the Parental Bonding Instrument “…Because that the data was non-normally distributed, we defined the total score of each dimension above and below the median score in this study as high and low child-rearing orientation, respectively. The median scores of care/affection, overprotection, and authoritarianism were 39, 14, and 12, respectively.” Please refer to line 173-177.

2.2.4. Child Behavior Checklist For Ages 6–18 “…We defined the T score of each dimension above and below the median T score in this study as high and low symptoms, respectively. The median scores of internalizing problems, externalizing, and ADHD symptoms were 61, 60, and 62, respectively.” Please refer to line 186-189.

2.2.5. Center for Epidemiological Study-Depression Scale ...We defined the total score above and below the median score of 12 in this study as high and low depressive symptoms, respectively. Please refer to line 198-200.

2.3. Statistical Analysis “…The predictive effects of caregivers’ demographic characteristics, depressive symptom, high orientation of stress-coping strategies, and child-rearing styles, and children’s demographic characteristics, and severe behavioral and ADHD symptoms (independent variables) on caregivers’ high affiliate stigma (dependent variable) were firstly examined by using univariate logistic regression analysis. Variables that significantly increased or decreased the risk of high affiliate stigma in univariate logistic regression analysis were further selected for multivariate logistic regression analysis, serving as the independent variables. The prediction of various coping orientations and parental bonding for affiliate stigma were examined separately in two logistic regression models, controlling for demographic characteristics, caregivers’ depressive symptoms, and the children’s behavioral and ADHD symptoms. Odds ratio (OR), its 95% confidence interval (CI), and a two-tailed p value of <0.05 were used to present statistical significance.” Please refer to line 206-219.

Comment 9

Can authors comment about effect sizes and indicate those in the statistical analysis section 2.3? Can authors also please add the effect sizes belonging to the regression analyses in Table 2?

Response

Thank you for your suggestion. Because the method of statistical analysis was changed to logistic regression analysis, we used odds ratio (OR) and its 95% confidence interval (CI) to present the effect sizes in the analysis. We added the method to define effect sizes as below. We also present OR and 95% CI in Tables 2 and 3, as well as in Results section as below.

We defined the OR that was higher than 1.5, 2.5 and 4 or lower than 0.667, 0.4 and 0.25 as to have a small, medium, and large effect respectively [29,30]. Please refer to line 219-221.

“All OR values indicated a small effect.” Please refer to line 257-258.

Comment 10

The effect of emotion-focused coping would not significant anymore after Bonferroni correction for multiple testing, authors could note this, since the sample size was not that small leaving enough power. Clearly, the effect for ‘less effective coping’ was more convincing.

Response

Thank you for your comment. We agreed the reviewer’s Comment 14 that “emotion focused coping can be positive or negative.” Therefore, we reorganized “emotion-focused coping” into seeking social support, changing cognition, and passive emotional coping. We also reanalyzed the prediction of stress-coping strategies according to new categories for high affiliate stigma and found that only seeking of social support significantly predicted the risk of high affiliate stigma. We rewrote the content of Results section as below.

The present study identified the following 5 subscales: problem-solving (including active coping, planning, suppression of competing activities, and restraint coping; Cronbach’s α = 0.936; median = 3.125), seeking social support (including seeking of instrumental and emotional social support; Cronbach’s α = 0.924; median = 3), changing cognition (including positive reinterpretation and acceptance; Cronbach’s α = 0.884; median = 3.25), passive emotional coping (including focusing on and venting of emotions and turning to religion; Cronbach’s α = 0.865; median = 2.25), and less effective coping (including denial, and behavioral and mental disengagement; Cronbach’s α = 0.849; median = 1.583). Because that the data was non-normally distributed, we defined the total score of each subscale above and below the median score in this study as high and low coping orientation, respectively. Please refer to line 153-164.

Table 3 details the results of the multivariate logistic regression analysis examining the predictors of affiliate stigma. The results of Model I examining the prediction of stress-coping orientations indicated that after controlling for the covariates, high orientation of seeking social support decreased the risk of high affiliate stigma. Please refer to line 251-258.

Comment 11

It is worthwhile to mention that effects remained significant independent from internalizing and ADHD problem severity while adjusting for it in the multivariate models.

Response

We mentioned it in Results section of the revised manuscript. Please refer to line 249-251.

Caregivers’ gender and depressive symptoms and children’s internalizing, externalizing and ADHD symptoms were used as the covariates.

Comment 12

Discussion, 4. First sentence, please refer to ‘parenting styles’.

Response

Thank you for your reminding. We changed it into “parental childrearing styles”. Please refer to line 266.

Comment 13

First sentence under 4.1, the phrasing could be easier, e.g. relate it to lower stigma and higher stigma (rather than saying negatively and positively related); perhaps address which is adaptive and which non-adaptive coping. I see that examples are following.

Response

Thank you for your suggestion. We revised the sentence as below to show the direction of association. Please refer to line 276-278.

This study demonstrated that high orientation of seeking social support significantly decreased the risk of high ADHD-related affiliate stigma, indicating that seeking social support is an adaptive stress-coping strategy.

Comment 14

Emotion focused coping can be positive or negative. As such, I do not understand authors’ reasonings why caregivers may experience fewer negative emotions, can authors please address this (“Because negative emotions may contribute to people’s misconceptions of social clues in social interactions [28], caregivers adopting emotion-focused coping strategies may experience fewer negative emotions, perceive less ADHD-related stigmatizing attitudes being directed toward them, and experience less internalization of these negative attitudes.”).

Response

I agree that emotion focused coping can be positive or negative. As mentioned in the response to Comment 10, we reorganized “emotion-focused coping” into seeking social support, changing cognition, and passive emotional coping. We also rewrote the content of 4.1. Coping Orientations and Affiliate Stigma based on the new results of statistical analysis as below. Please refer to line 279-305.

“… The present study examined two dimensions of seeking social support, including instrumental and emotional social support. Seeking of instrumental social support such as learning from the experience of others may help the caregivers increases the opportunity to successfully manage stressors that they encounter. Seeking of emotional social support such as getting sympathy and understanding from others may diminish the negative emotional consequences of stressful events [8,9]. Social support may contribute to caregivers’ success in managing children’s ADHD-related problems and enhance caregivers’ emotional health. Social support may also reduce the risk of social withdrawal for caregivers of children with ADHD and increase the opportunity for people to understand caregivers’ care burden. Therefore, social support may reduce people’s ADHD-related stigmatizing attitudes toward the caregivers.

In addition to seeking socioemotional support, the coping strategy of changing cognition such as positive reinterpretation, acceptance of stressors, and perceived growth in the process of managing stress may diminish the negative emotional consequences of stressful events [8,9]. Because negative emotions may contribute to people’s misconceptions of social clues in social interactions [28], it is reasonable to hypothesize that caregivers adopting the coping strategy of changing cognition may experience less internalization of ADHD-related stigmatizing attitudes.表單的底部 However, the prediction of changing cognition for low affiliate stigma was found in univariate but not in multivariate logistic regression analysis.”

Comment 15

Section 4.1. latter part, would it not be advisable to summarize a separate section addressing possible clinical implications at the end of the discussion.

Response

We added a new paragraph titled as “4.3. Implication” as below into the revised manuscript. Please refer to line 329-340.

“4.3. Implication

Although the cross-sectional study design limited the possibility to determine the causal relationship between affiliate stigma, stress-coping orientations and parental child-rearing styles, mental health professionals must assist the caregivers of children with ADHD in identifying the most suitable stress-coping strategies, especially seeking of social support, and advising against less effective coping strategies. Moreover, intervention programs targeting caregiver affiliate stigma must consider the parental styles in their approach. Affiliate stigma, stress-coping strategies, and parental child-rearing styles may be the results of interactions between caregivers and environments. Not only caregivers’ psychological and cognitive mechanisms but also the interactions with people in caregivers’ microsystem and mesosystem should be assessed.

Comment 16

Section 4.2. Here, in the first sentence, authors repeat themselves again, again it could be phrased more attractively and less technical as in a results section. For example, “As may be expected, a positive parental style of care and affection was associated with experiencing less affiliate stigma….” Etcetera. In contrast, overprotection usually refers to a maladaptive or negative parenting style.

Response

Thank you for your suggestion. We revised this sentence as below. Please refer to line 307-309.

The study demonstrated that the parental style of care and affection decreased the risk of high affiliate stigma, whereas the overprotective style increased the risk of high affiliate stigma.

Comment 17

Regarding “Research has demonstrated that high levels of caregiver care and affection significantly correlate with reduced inattention and psychiatric comorbidities in their children [20], which may in turn reduce other people’s negative attitudes toward caregivers.”, on page 8, it may be worthwhile to mention that the reported association was independent from ADHD symptom severity. All in all, authors may also mention that a generally more healthy psychological functioning may be expressed in all kinds of better psychological measures to which affiliate stigma may also belong, in other words, a happier person will score better on different kinds of measures reflecting psychological well-being, and vice versa, a person with more maladaptive coping and parenting strategies will also be more susceptible to experiencing affiliate stigma.

Response

Thank you for your suggestion. We revised the paragraph based on your suggestion as below. Please refer to line 309-321.

Research has demonstrated that high levels of caregiver care and affection significantly correlate with reduced inattention and psychiatric comorbidities in their children [20]. Charbonnier et al. demonstrated that children’s ADHD symptoms were positively associated with affiliate stigma [3]. Therefore, the parental style of care and affection may decrease the risk of high affiliate stigma. However, the present study found that the parental style of care and affection decreased the risk of high affiliate stigma after controlling for the effects of children’s ADHD symptoms. The result indicated that the association between the parental style of care and affection and low affiliate stigma may not be fully accounted for the reduced ADHD symptoms. It is possible that both the parental style of care and affection and low affiliate stigma resulted from caregivers’ healthy psychological functioning. A happier person may score higher on various kinds of measures reflecting psychological well-being.

Comment 18

Section 4.2, I assume the authors refer to their own study? Then better say: “the results of this study indicate”

Response

We revise it into “The results of this study”. Please refer to line 307.

Comment 19

Conclusions: Again here, better avoid too technical language. It reads much like a reiteration of results. Wording is quite repetitive. Further, what suggestions for further research can authors give us?

Response

We revised the content of conclusion section as below based on your suggestion. Please refer to line 350-364.

This study demonstrated that seeking social support may be an adaptive stress-coping strategy to protect caregivers of children with ADHD from aggravated affiliate stigma. Moreover, adopting the parental style of care and affection may contribute to low affiliate stigma, whereas adopting the parental style of overprotection may increase the risk for caregivers to perceive public stigma and internalize the negative attitudes. The results of this study indicate that caregivers of children with ADHD need help to develop the coping strategy of seeking social support. Intervention programs targeting caregiver affiliate stigma must consider the parental styles in their approach. Further study is needed to examine the mechanism accounting for the associations of seeking social support and the parental style of care and affection and overprotection with affiliate stigma in caregivers of children with ADHD. Mental health professionals should develop the intervention programs to enhance caregivers’ adaptive stress-coping strategies and parental child-rearing styles; the effects of intervention programs on reducing affiliate stigma also warrants study.”